# Comparison of Aneurysm Patency and Mural Inflammation in an Arterial Rabbit Sidewall and Bifurcation Aneurysm Model under Consideration of Different Wall Conditions

**DOI:** 10.3390/brainsci10040197

**Published:** 2020-03-27

**Authors:** Basil Erwin Grüter, Stefan Wanderer, Fabio Strange, Sivani Sivanrupan, Michael von Gunten, Hans Rudolf Widmer, Daniel Coluccia, Lukas Andereggen, Javier Fandino, Serge Marbacher

**Affiliations:** 1Department of Neurosurgery, Kantonsspital Aarau, 5000 Aarau, Switzerland; stefan.wanderer@ksa.ch (S.W.); Fabio.Strange@ksa.ch (F.S.); daniel.coluccia@luks.ch (D.C.); lukas.andereggen@ksa.ch (L.A.); javier.fandino@ksa.ch (J.F.); serge.marbacher@ksa.ch (S.M.); 2Cerebrovascular Research Group, Neurosurgery, Department of BioMedical Research, University of Bern, 3010 Bern, Switzerland; sivani.sivanrupan@students.unibe.ch; 3Institute of Pathology Laenggasse, 3063 Ittigen, Switzerland; mvongunten@patholaenggasse.ch; 4Department of Neurosurgery, Neurocenter and Regenerative Neuroscience Cluster, Inseslspital, Bern University Hospital, University of Bern, 3010 Bern, Switzerland; hansrudolf.widmer@insel.ch

**Keywords:** aneurysm, decellularization, inflammation, rabbits, vessel wall

## Abstract

*Background:* Biological processes that lead to aneurysm formation, growth and rupture are insufficiently understood. Vessel wall inflammation and degeneration are suggested to be the driving factors. In this study, we aimed to investigate the natural course of vital (non-decellularized) and decellularized aneurysms in a rabbit sidewall and bifurcation model. *Methods:* Arterial pouches were sutured end-to-side on the carotid artery of New Zealand White rabbits (vital [*n* = 6] or decellularized [*n* = 6]), and into an end-to-side common carotid artery bifurcation (vital [*n* = 6] and decellularized [*n* = 6]). Patency was confirmed by fluorescence angiography. After 28 days, all animals underwent magnetic resonance and fluorescence angiography followed by aneurysm harvesting for macroscopic and histological evaluation. *Results:* None of the aneurysms ruptured during follow-up. All sidewall aneurysms thrombosed with histological inferior thrombus organization observed in decellularized compared to vital aneurysms. In the bifurcation model, half of all decellularized aneurysms thrombosed whereas the non-decellularized aneurysms remained patent with relevant increase in size compared to baseline. *Conclusions:* Poor thrombus organization in decellularized sidewall aneurysms confirmed the important role of mural cells in aneurysm healing after thrombus formation. Several factors such as restriction by neck tissue, small dimensions and hemodynamics may have prevented aneurysm growth despite pronounced inflammation in decellularized aneurysms. In the bifurcation model, rarefication of mural cells did not increase the risk of aneurysm growth but tendency to spontaneous thrombosis.

## 1. Introduction

In intracerebral aneurysms, the risk of growth and rupture is associated with larger aneurysm size, larger aneurysm height to neck aspect ratio and irregular configuration of the aneurysm [1,2,3]. However, the biological mechanisms of these morphological characteristics are poorly understood. There is a growing body of evidence that chronic vessel wall inflammation and loss of aneurysm mural cells is a crucial factor in the pathogenesis of aneurysm growth and rupture [4,5,6]. Aneurysms with vital vessel walls may be able to recruit smooth muscle cells that are able to organize thrombus, to build a neointima and, by phenotype switch, to synthesize extracellular matrix. On the other hand, those aneurysms with a rarefication of cells in their vessel wall seem to be unable to promote aneurysm healing after intraluminal thrombosis. Instead, intra-aneurysmal thrombus may promote chronic inflammation, further weakening of the vessel wall and finally leading to aneurysm growth and rupture [6,7]. This difference becomes fundamentally crucial with endovascular aneurysm treatments, which are conceptually based on a biological healing process, rather than just mechanical flow obstruction [7,8,9].

The abovementioned putative pathophysiological mechanism was first observed in human samples [10] and later confirmed in an experimental setting in rat saccular sidewall aneurysms [11]. Rabbits stand higher up in the translational chain than rats and allow for experimental creation of complex, more physiological bifurcation aneurysms by means of rheology and hemodynamics [12,13]. Rabbit models are considered ideal for testing of novel endovascular devices, because the rabbit carotid artery is accessible with endovascular devices of the same size as in humans. Therefore, this study aims to investigate the natural course of vital and decellularized aneurysms in a rabbit sidewall and bifurcation aneurysm model with an emphasis on aneurysm patency, growth and mural inflammation.

## 2. Materials and Methods

New Zealand white rabbits aged 4 months (weighing 3750 ± 293 g) received care in accordance with institutional guidelines. The Committee for Animal Care of the Canton Bern, Switzerland (BE 108/16) approved the experiments. An a priori power analysis was performed, revealing *n* = 6 animals per group needed to reach statistical significance with an estimation of 30% difference between groups. Two animals served as pilots. All animals were randomly allocated to either vital or decellularized aneurysm group. For each group, 6 aneurysms were created. For sidewall aneurysm creation, two animals were used as tissue donors. Two aneurysms (one on each common carotid artery) were created in one animal. For bifurcation aneurysms, only one aneurysm was created per animal. Graft interpositions were taken from the same animal, with no need for additional donor animals.

### 2.1. Creation of Sidewall Aneurysms

Female rabbits were premedicated with an intramuscular injection of Ketamine HCL 30 mg/kg (Ketalar, 50 mg/mL, Pfizer AG, Zürich Switzerland) and Xylazine 6 mg/kg (Xylapan 20 mg/mL). An auricular vein was then catheterized and a continuous infusion of anesthesia solution (10mL Ketalar and 1.6 mL Xylapan in 50 mL NaCl) was installed with a flow rate of 4–14 mL/h. Furthermore, Fentanyl 1 mg/kg (Fentanyl, Janssen-Cilag, Zug, Switzerland) was applied for analgesia. Animals breathed spontaneously through an oxygen mask. During the operation, animals were located on a heating panel and physiological variables such as heart rate, blood pressure and temperature were continuously monitored. After local infiltration of the pectoral musculature with lidocaine (Lidocaine 1%, Streuli & Co, Uznach, Switzerland), the common carotid artery was dissected on both sides and a previously prepared donor graft (either vital or decellularized) was sutured in an end-to-side configuration, to form a sidewall aneurysm. Nimodipine (Nimotop 0.2 mg/mL, Bayer, Leverkusen, Germany) was locally applied to prevent for vasospasms. A fluorescence angiography was then performed [14,15], to ascertain aneurysm perfusion and patency of the underlying vessel. Afterwards, incised tissues (musculature, subcutaneous and skin) were readapted and closed. Postoperative analgesia was ascertained with transdermal fentanyl application (12 μg/72 h). All animals received postoperative antibiotics by intramuscular injection of terramycin (60 mg/kg), vitamin B12 (Novartis, Basel, Switzerland) 100 mcg subcutaneous and prophylactic low-molecular weight heparin (250 units/kg) subcutaneous.

### 2.2. Creation of Bifurcation Aneurysm

Due to an internal periodic veterinarian re-evaluation of the standards, anesthesia protocols were slightly adopted for bifurcation models. Premedication comprised subcutaneous application of Ketamine 20 mg/kg, Dexmedetomidine (Novartis, Basel, Switzerland) 100 mg/kg and Methadone (Novartis, Basel, Switzerland) 0.3 mg/kg. Animals were then preoxygenated through a facial mask, before installation of peripheral catheters in the auricular vein and in the contralateral auricular artery. Then, propofol (1–5 mg/kg) (Novartis, Basel, Switzerland) and 0.2–1 mg/kg midazolam (Novartis, Basel, Switzerland) were intravenously administered, followed by intubation with an endotracheal tube (3 mm). Mode of the breathing system was chosen circle, able to be changed from ventilation to spontaneous breathing anytime. A heating pad was continuously used to keep the animals warm during the procedure. For monitoring, a continuous electrocardiogram, a rectal temperature probe and a bispectral index were installed. Analgesia was ascertained by local subcutaneous infiltration with ropivacaine (Novartis, Basel, Switzerland), followed by a continuous rate of infusion of 50 mcg/kg/min lidocaine (Novartis, Basel, Switzerland) and fentanyl boli of 3–10 mcg/kg/h. Postoperatively, Meloxicam 0.5 mg/kg (Novartis, Basel, Switzerland), Vitamin B12 100 mcg (Novartis, Basel, Switzerland) and Clamoxyl 20 mg/kg (Novartis, Basel, Switzerland) were administered subcutaneously. For the first three days, low-molecular weight heparin (250 units/kg) and meloxicam were administered subcutaneously (likewise methadone was administered, if an additive was needed). The detailed surgical technique for creation of bifurcation aneurysms has been presented elsewhere [16]. Briefly, bifurcation aneurysms were created by end-to-side anastomosis of the right common carotid artery to the left common carotid artery and interposition of an arterial pouch, either vital or decellularized.

### 2.3. A Protocol for Decellularization

Untreated donor arterial grafts with a standardized length of 3–4 mm were taken from a segment of the common carotid artery of a donor animal, ligated with a 6-0 suture on one end and immediately reimplanted in a recipient animal or stored in phosphate buffered saline (PBS) at −4 °C for a maximum of 3 days. All aneurysm pouches were measured, and photo documented on creation and again at follow-up. For decellularization, a modified protocol of a previously described methodology was performed [11,17]: First, grafts were frozen in PBS at −4 °C for several days. Later, they were thawed, rinsed with distilled water and incubated in 1% sodium dodecyl sulphate (SDS) for 6 h at room temperature. The SDS-treated grafts were then washed with gentle shaking and refrozen and kept in PBS at −4 °C until reimplantation. To establish these modifications of the original protocol, various SDS concentration (0.1% and 1%) and several time spans for decellularization (6 h, 9 h,12 h, 15 h and 2 h, 4 h, 6 h, 8 h, respectively) were assessed. All samples were histologically cut and stained with 4′,6-diamidino-2-phenylindole (DAPI) to count nuclei and with hematoxylin-eosin (HE), to assess the integrity of the extracellular matrix such as elastic fibers. Cell nuclei were counted three times for three randomly selected cuts, in each slice specifically for the following wall layers of the vessel: endothelium, media and adventitia. Digital photographs of the microscopic images were taken and analyzed while blinded to the treatment. Near-complete graft decellularization with extracellular fibers still intact was documented after 6 h of 1% SDS treatment.

### 2.4. Outcome Measurements

After creation, sidewall aneurysms were followed with color coded duplex sonography (SonoSite 180 PLUS, SonoSite, Bothell, WA, USA) on post-operative day 1, day 3 and every 7 days thereof. After a follow-up period of 28 days all animals underwent MRI with MR-angiography (MRA). Immediately afterwards aneurysms were surgically re-exposed, and a fluorescence angiography was performed before euthanasia with an overdose of thiopental (Esconarkon ad us. vet, Streuli & Co, Uznach, Switzerland) and tissue harvesting. Aneurysms were macroscopically inspected and measured. Aneurysm volume was calculated on the basis of a = length, b = width and l = height, with the formula π(1.5(a + b) − $\sqrt{ab}$)(l − b) + (2/3 × π × ab^2^). Afterwards fixation in formalin (4% weight/volume solution, J.T. Baker, Arnhem, The Netherlands) and embedding in paraffin for histological analysis followed. Histological staining included HE, Masson–Goldner trichrome, smooth muscle actin, and von Willebrand factor (F8) staining. Stained slices were digitalized (omnyx VL120, GE healthcare, Chicago, IL, USA) and evaluated with the JVS viewer (JVS view 1.2 full version, University of Tampere, Finland). Histologic scoring was performed blinded to treatment allocation. A 4-scale grading system (“none”, “mild”, “moderate”, “severe”) was applied to characterize histology, according to the previously presented neointima score [11].

### 2.5. Statistics

Data were analyzed and visualized using Graph Pad Prism statistical software 8.3.1 for Windows (GraphPad Software, Sand Diego, CA, USA). Unpaired Mann–Whitney test was used to calculate differences between vital and decellularized aneurysms according to histological analysis with non-parametric values. Values are presented as median with interquartile range and arbitrary units 0–3 representing categories (“none”, “mild”, “moderate”, “severe”) according to the neointima score [11]. A *p*-value of <0.05 was considered statistically significant and a *p*-value of <0.001 was considered highly significant.

## 3. Results

During the study period, no aneurysm ruptured. All sidewall aneurysms (vital and decellularized) thrombosed spontaneously during follow-up. Histologically, inferior thrombus organization was observed in decellularized aneurysms when compared to healing characteristics in vital aneurysms. In the arterial pouch bifurcation model, three out of six aneurysms with decellularized walls thrombosed spontaneously whereas all vital aneurysms (six out of six) stayed patent, with relevant growth pattern in two cases.

### 3.1. Study and Animal Characteristics

Totally, 22 New Zealand white rabbits were included in this study, weighting 3750 ± 293 g. No animals had to be excluded due to severe comorbidities and no animal died prematurely before planed euthanasia on follow-up day 28. See Figure 1 for an overview of the experimental setting.

For histological evaluation one vital aneurysm in the sidewall constellation and one decellularized aneurysm in the bifurcation constellation was excluded from the final analysis due to insufficient detection of the relevant structures after histologic processing of these heavily scarred aneurysms.

### 3.2. Aneurysm Patency

All sidewall aneurysms showed initial flow upon creation but thrombosed within the first two weeks after creation and were not detectable thereafter with either ultrasound or MR angiography. Intraoperative fluorescence angiography confirmed flow obliteration in all these cases. Calculated volume, based on the measured aneurysm size, was significantly smaller for scarred aneurysms at follow-up (7.55 ± 2.73 mm^3^) than they were at creation (11.27 ± 3.27 mm^3^), *p* = 0.0033 (Figure 2).

In bifurcation aneurysms, only three out of six aneurysms with decellularized walls thrombosed and all of these with vital vessel walls remained patent until follow-up (exemplary illustration in Figure 3).

Furthermore, these aneurysms showed a pattern of growth from (6.48 ± 1.81 mm^3^) on creation to (19.48 mm^3^ ± 6.40 mm^3^) follow-up (*p* = 0.037)

### 3.3. Histological Analyses

Overall, there was more inflammation in decellularized aneurysms than in those with vital vessel walls. In sidewall aneurysms, this was reflected by significantly more neutrophil invasion in the thrombus in decellularized than in vital aneurysms (*p* = 0.0065) (Figure 4).

In bifurcation aneurysm, there were significantly more inflammatory cells (neutrophils) in the wall of decellularized aneurysms compared to vital aneurysms (*p* = 0.013). Periadventitional fibrosis was higher in vital aneurysms than in decellularized ones (*p* = 0.013). All histological characteristics are summarized in Figure 5.

## 4. Discussion

The results of this study demonstrate that all arterial pouch (decellularized and non-decellularized) sidewall aneurysms thrombose spontaneously during follow-up without increase in size. Poor thrombus organization in decellularized sidewall aneurysms confirms the important role of mural cells in aneurysm healing. In the bifurcation aneurysm model, removal of mural cells did not increase the risk of aneurysm growth.

In our experiments, all sidewall aneurysms thrombosed spontaneously without any treatment. This is opposed to the natural course of saccular sidewall aneurysms which were sutured as standardized arterial pouches on the abdominal aorta in a rat model [18]. In that model the authors found a clear pattern of growth in decellularized aneurysms [11]. We hypothesize that the pressure of surrounding muscular tissues in the rabbit neck may counteract the artificially created saccular aneurysms from growth. Furthermore, the base dimensions of these aneurysms are given by the diameter of the carotid artery of the donor animal. This size was usually smaller (approximately 1–1.5 mm) than in a rat aorta (2–3 mm). Together with the different hemodynamics between the rat aorta and the rabbit carotid artery, these aneurysms may have been simply too small for a sufficient perfusion, particularly since the relatively thick and muscular arterial walls may have a tendency to self-contract or increased fibrosis after transplantation. Ding et al. found a patency rate of 95% after 3 weeks in venous pouch sidewall aneurysms on rabbit carotids [19].

In order to overcome these limitation factors, as a next step we repeated the series with non-decellularized and decellularized arterial pouch aneurysms in a hemodynamically more challenging bifurcation constellation. Previous experiments demonstrated in various species (rats, rabbits and dogs) that spontaneous thrombosis occurs less frequently in bifurcation than sidewall venous pouch aneurysms [20,21,22]. In contrast to these earlier findings, however, all decellularized arterial pouch aneurysms thrombosed even in the setting of an artificial bifurcation. Most previous studies with degenerated vessel walls used elastase eradication of the cells [23,24,25,26]. Sodium dodecyl sulfate (SDS) is a detergent that destroys cells but leaves extracellular matrix intact. Its use for experimental decellularization worked well in a previously established rat model, where decellularized aneurysms have been shown to grow over time and eventually rupture, in contrast to aneurysms with vital vessel walls [6,9,11]. However, the completely decellularized graft (including eradication of endothelial cells) after SDS treatment may exhibit prothrombogenic properties. This may be an explanation for the finding of high rate of thrombosed decellularized bifurcation aneurysms. The growing pattern of bifurcation aneurysms with vital vessel walls indicates that the hemodynamic constellation (in a vessel bifurcation) is an important factor for aneurysm enlargement/growth.

When comparing vital and decellularized aneurysms histologically, there was a clear pattern of more pronounced inflammation in decellularized aneurysms for both sidewall and bifurcation aneurysms. This is in line with previous findings [27,28,29]. For aneurysm healing, intraluminal thrombus needs to undergo gradual organization into a mature thrombus and a neointima needs to form. This process is mediated by smooth muscle cells and myofibroblasts, which migrate into the thrombus, presumably originating in the vessel wall. If there is a substantial diminution in the pool of these cells (i.e., after decellularization) the intraluminal thrombus will undergo cycles of lysis and re-thrombosis instead of scarification [6,11]. This instable thrombus formation causes local inflammatory reactions which promotes further vessel wall weakening.

In summary, there was more pronounced inflammation in decellularized aneurysm than in those with vital walls. However, decellularized aneurysms did not show any pattern of growth or rupture, neither in a sidewall nor in a bifurcation constellation. Therefore, the presented aneurysm models need further refinements to allow for meaningful experiments with a translational focus. Further experiments should use other degrading substances like elastase, test anti-platelet medications to prevent spontaneous thrombus formation, or relocation of the experimental aneurysms into the abdominal cavity to allow for more unrestricted growth. However, with all these issues addressed, still no animal model ever will perfectly match all aspects of the human condition of the disease [30,31]. For instance, there are relevant differences in thrombus formation and endothelial cell coverage between rabbits and humans. In addition, we used healthy arteries in which cells but not the extracellular matrix was destroyed to form aneurysms. However, the elastin content of real aneurysms is inferior to that of healthy arteries [32]. Furthermore, the aneurysm angioarchitecture influences hemodynamic characteristics, and with that the rate of spontaneous thrombosis. Despite all efforts made to standardize aneurysm dimensions and geometry (the latter specifically for either sidewall or bifurcation constellation), we could not avoid differences of few millimeters in size of the vessel pouches and thus in hemodynamics. Further limitations include the relatively small sample size of *n* = 6 animals per group. Lastly, a longer follow-up than 28 days would probably be better to characterize hemodynamic-induced changes in the bifurcation constellation.

## 5. Conclusions

The results of poor thrombus organization in decellularized rabbit arterial sidewall aneurysms confirm the important role of mural cells in aneurysm healing after intraluminal thrombus formation. Several factors such as restriction by neck tissue, small dimensions and hemodynamics may have prevented aneurysm growth despite pronounced inflammation in decellularized aneurysms. Even in the bifurcation aneurysm model, removal of mural cells did not increase the risk for aneurysm growth but resulted in a higher rate of spontaneous thrombosis. Future studies should examine the role of less thrombogenic degenerated aneurysm wall pouches in a rabbit artificial bifurcation model.

## Figures and Tables

**Figure 1 brainsci-10-00197-f001:**
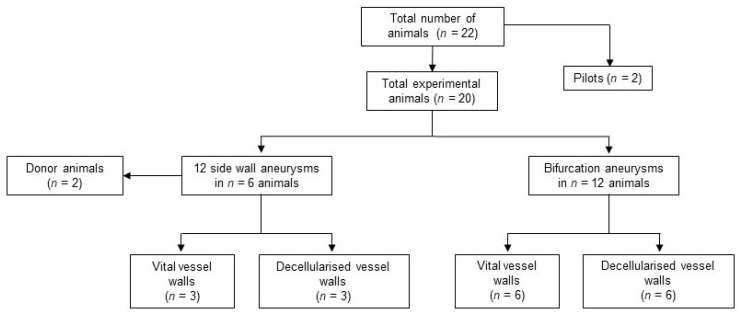
Study design and animal numbers. No animals had to be excluded prematurely for morbidity or mortality.

**Figure 2 brainsci-10-00197-f002:**
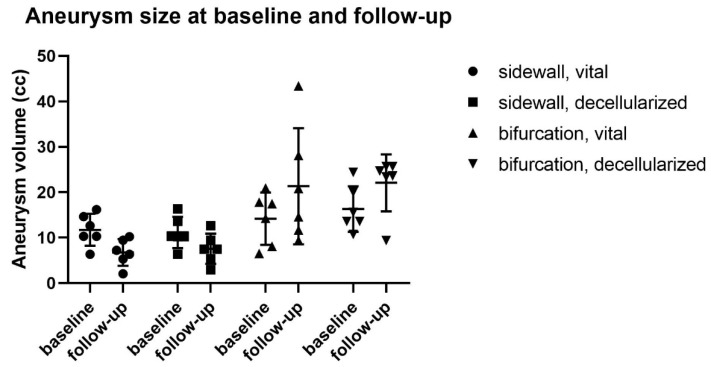
Aneurysm size at baseline and follow-up. Relevant growth pattern was observed in 2 cases of vital bifurcation aneurysms, whereas all the sidewall aneurysms thrombosed spontaneously.

**Figure 3 brainsci-10-00197-f003:**
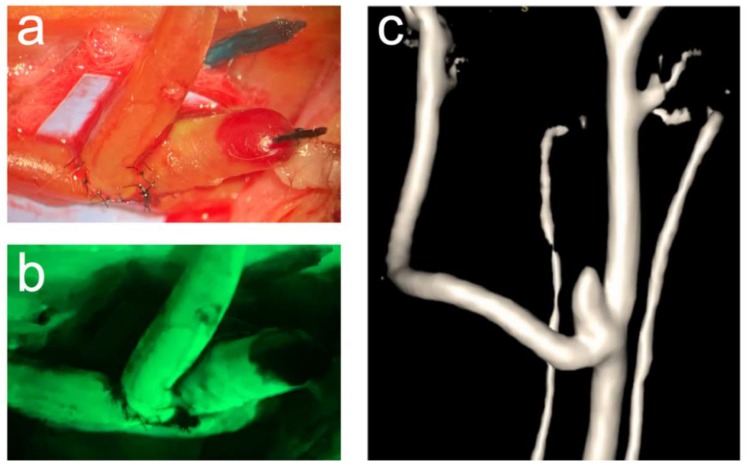
Exemplary illustration of a patent vital bifurcation aneurysm. The operative situs through the operative microscope (a) and the corresponding fluorescence angiography (**b**) visualize blood flow in both, the aneurysm and the parent artery at the time of aneurysm creation. One-month patency is confirmed by magnetic resonance angiography (**c**).

**Figure 4 brainsci-10-00197-f004:**
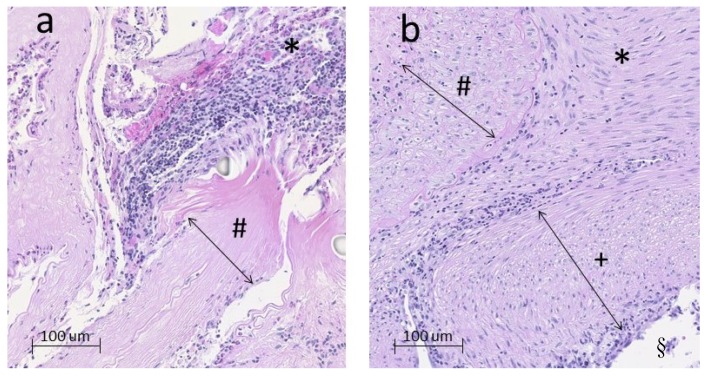
Exemplary histology on a 16-fold digital zoom of a decellularized (**a**) and a vital (**b**) aneurysm in sidewall constellation. The degenerated aneurysm wall (# in a) contains predominantly extracellular matrix fibres only. By contrast, the vital aneurysm wall (# in b) is marked by a high cell density. Inside the thrombus (*) of decellularized aneurysms (**a**), excessive neutrophil infiltration was observed. In (**b**), hardly any neutrophils are visible and derivates of myofibroblast have organized the former intraluminal hematoma into mature thrombus and scare tissue. Furthermore, a thick and consistent neointima (+) separates the former aneurysm cavity from the lumen of the parent artery (§) (similar but no visible in (**a**)).

**Figure 5 brainsci-10-00197-f005:**
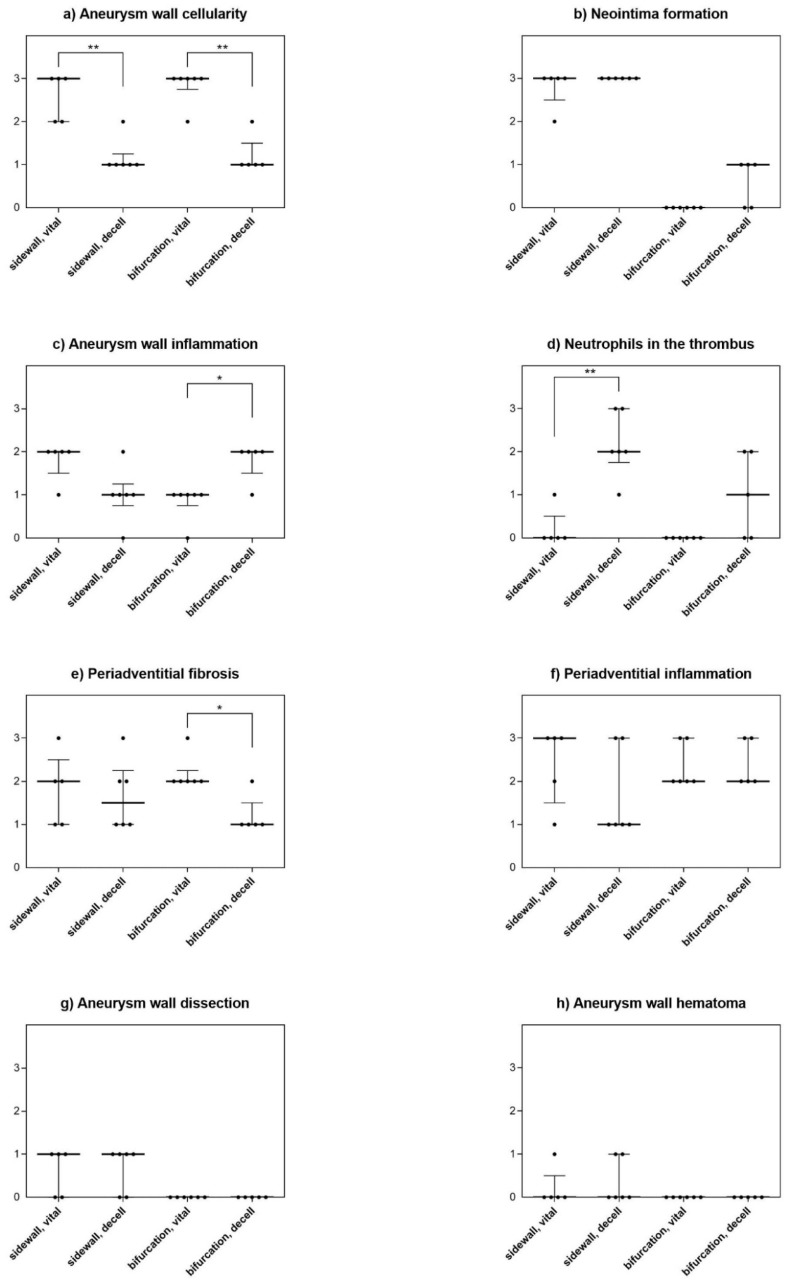
Detailed histological findings for all analyzed features. In both models, aneurysm wall cellularity was significantly lower in decellularized aneurysms than in vital aneurysms, confirming a successful experimental decellularization (**a**). Spontaneous thrombosis and neointima formation (**b**) were strong in the sidewall constellation, but not so in the bifurcation model. In the bifurcation model, aneurysm wall inflammation was significantly more pronounced in decellularized aneurysms when compared with vital aneurysms (**c**). However, there was no difference in terms of aneurysm wall inflammation in the sidewall model. On the other hand, there were significantly more inflammation cells, i.e., neutrophils in the thrombus of decellularized sidewall aneurysms, a difference not as distinctly observed in the bifurcation constellation (**d**). In turn, periadventitial fibrosis was significantly higher in vital than in decellularized bifurcation aneurysms, but not in sidewall aneurysms (**e**). There were no relevant differences for periadventitial inflammation (**f**), aneurysm wall dissection (**g**) or aneurysm wall hematoma (**h**) between different wall conditions for either aneurysm model. A 4-scale grading system 0 = none, 1 = mild, 2 = moderate, 3 = severe was applied to characterize histology [11]. *: *p* < 0.05, ***p* < 0.001.

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
