# Peer review of "Comparison of Aneurysm Patency and Mural Inflammation in an Arterial Rabbit Sidewall and Bifurcation Aneurysm Model under Consideration of Different Wall Conditions"

_brainsci, 2020, doi:10.3390/brainsci10040197_

Round 1

Reviewer 1 Report

This study by Grüter et al, evaluates the natural course of vital (non-decellularized) and decellularized aneurysms in a rabbit sidewall and bifurcation model. The hypothesis is interesting. Nevertheless, I have some comments.

1-Please, show a graph regarding aneurysm size at baseline and final follow up (individual data points).

2- In my view, it would be interesting to show data regarding angiogenesis (Cd31+ microvessels) and metalloprotease (MMP9) expression in non-decellularized and decellularized aneurysms.

3-Please, authors should justify why they use female rabbits instead of male for these studies. Is there any difference in the development of aneurysm between both sexes?

Reviewer 2 Report

Review of: Comparison of Aneurysm Patency and Mural 2 Inflammation in an Arterial Rabbit Sidewall and 3 Bifurcation Aneurysm Model under Consideration of 4 Different Wall Conditions

This work addresses the important challenge of developing an animal model that is suitable for studying the degenerative changes in the walls of human cerebral aneurysms.    In this work, arterial pouch saccular aneurysms in rabbits were created in carotid arteries of New Zealand white rabbits.  Variations included sidewall versus bifurcation and cellularized versus decellularized arterial tissue.  The focus on cellularized versus decellularized walls comes from the conjecture that aneurysms with few intramural cells will be unable to recruit additional cells to the wall and will be less able to remodel after thrombosis, instead enduring prolonged inflammation that may render the wall vulnerable to rupture.

The focus of this work is important and the team has included imaging of the lumen to assess thrombosis as well as histological evaluation of the tissue at 28 days to study the connection between thrombus formation, cellularization and wall changes.   There are some major concerns about the study design and relevance of the work to human aneurysms that reduce enthusiasm for the work. 

1) The most serious concern is the inconsistency in methodology for the sidewall and bifurcation groups.  One difference seems to be the use of heparin and meoxicam in the bifurcation protocol that was not reported  in the sidewall protocol. “For the first three days, low-molecular weight 108 heparin (250 units/kg) and meloxicam were administered subcutaneously (likewise methadone was 109 administered, if an additive was needed). “  This difference is quite important as the sidewall aneurysms were reported to be completely occlude by thrombus in contrast to the bifurcation aneurysms.  While there could be other explanations that are flow related, the apparent difference in protocol is so substantial as to render any comparison to be of questionable value. 

For example- the following text compares these groups:

Spontaneous  thrombosis and neointima formation (b) were strong in the sidewall constellation, but not so in the  bifurcation model. On the other hand, there were significantly more inflammation cells, i.e. neutrophils in the thrombus of decellularized  sidewall aneurysms - a difference not as distinctly observed in the bifurcation constellation

2) A second substantial concern is the relevance of this particular model to humans.  It is clear that no animal model will perfectly match all aspects of the human condition. However, it is important that the model does have relevance for the focus of the study. In this particular case, the focus is on thrombus formation and inflammation.   However, the relevance of a model aneurysm that fully thromboses within two weeks is not clearly made.    

3) Some limitations were well addressed in the Discussion, though there were some importation limitations that still need to be addressed. For example, in addition to the comments above, differences between rabbit and humans with respect to thrombus formation and endothelial cell coverage were not discussed.   It is not clear how much variation there was in the geometry of the pouch when it was initially created.    Sample size limitations were not considered.  It would have been better to focus on one geometry and have more cases.  Differences between walls of arteries and aneurysm walls are also not discussed sufficiently.   The elastic properties are extremely different as is the elastin content.

Round 2

Reviewer 1 Report

No more comments